# Ribosomal DNA copy number is associated with body mass in humans and other mammals

Pui Pik Law[1,2,7], Liudmila A. Mikheeva[1,7], Francisco Rodriguez-Algarra[2,7], Fredrika Asenius[3], Maria Gregori[3], Robert A. E. Seaborne[2,6], Selin Yildizoglu[2], James R. C. Miller[1], Hemanth Tummala[2], Robin Mesnage[1], Michael N. Antoniou[1], Weilong Li[4], Qihua Tan[5], Sara L. Hillman[3], Vardhman K. Rakyan[2], David J. Williams[3] & Michelle L. Holland[1] ✉

Body mass results from a complex interplay between genetics and environment. Previous studies of the genetic contribution to body mass have excluded repetitive regions due to the technical limitations of platforms used for population scale studies. Here we apply genome-wide approaches, identifying an association between adult body mass and the copy number (CN) of 47S-ribosomal DNA (rDNA). rDNA codes for the 18 S, 5.8 S and 28 S ribosomal RNA (rRNA) components of the ribosome. In mammals, there are hundreds of copies of these genes. Inter-individual variation in the rDNA CN has not previously been associated with a mammalian phenotype. Here, we show that rDNA CN variation associates with post-pubertal growth rate in rats and body mass index in adult humans. rDNA CN is not associated with rRNA transcription rates in adult tissues, suggesting the mechanistic link occurs earlier in development. This aligns with the observation that the association emerges by early adulthood.

Lifestyle changes have driven a relentless increase in the incidence of obesity[1]. Current interventions have proven insufficient to curb this trend, therefore, understanding the basis of an individual's response to an obesogenic environment is of great interest. Genome-wide association studies (GWAS) have contributed to our understanding of the genetic influence over body mass, yet so far only part of the heritability estimated by family and twin studies can be explained[2,3]. Epigenetic mechanisms have also been explored in depth but largely using array technologies that only partially capture the DNA methylome[4]. Due to technical limitations, repetitive parts of the genome, such as 47S-ribosomal DNA (rDNA) have been excluded from such studies[5].

Here we utilize whole-genome approaches, inclusive of repetitive regions to identify an association between rDNA copy number (CN) and body mass in humans and rats. We observe this association in two independent cohorts and different tissues in adult humans. The association is not driven by cell-type specific variation in rDNA CN and rDNA CN is not altered between monozygotic twins discordant for body mass index (BMI), implying that the association is not downstream of environmental exposures or metabolic changes that occur with BMI variation. DNA methylation at the rDNA in adult tissues is correlated with the rDNA CN and acts to normalize the transcription rates. Therefore, if the mechanistic basis of the association is through

[1]Department of Medical and Molecular Genetics, School of Basic and Medical Biosciences, King's College London, London, UK. [2]The Blizard Institute, School of Medicine and Dentistry, Queen Mary University of London, London, UK. [3]UCL EGA Institute for Women's Health, University College London, London, UK. [4]Population Research Unit, University of Helsinki, Helsinki, Finland. [5]Epidemiology, Biostatistics and Biodemography, Department of Public Health, University of Southern Denmark, Copenhagen, Denmark. [6]Present address: Centre for Human and Applied Physiological Studies, King's College London, London, UK. [7]These authors contributed equally: Pui Pik Law, Liudmila A. Mikheeva, Francisco Rodriguez-Algarra. ✉e-mail: michelle.holland@kcl.ac.uk

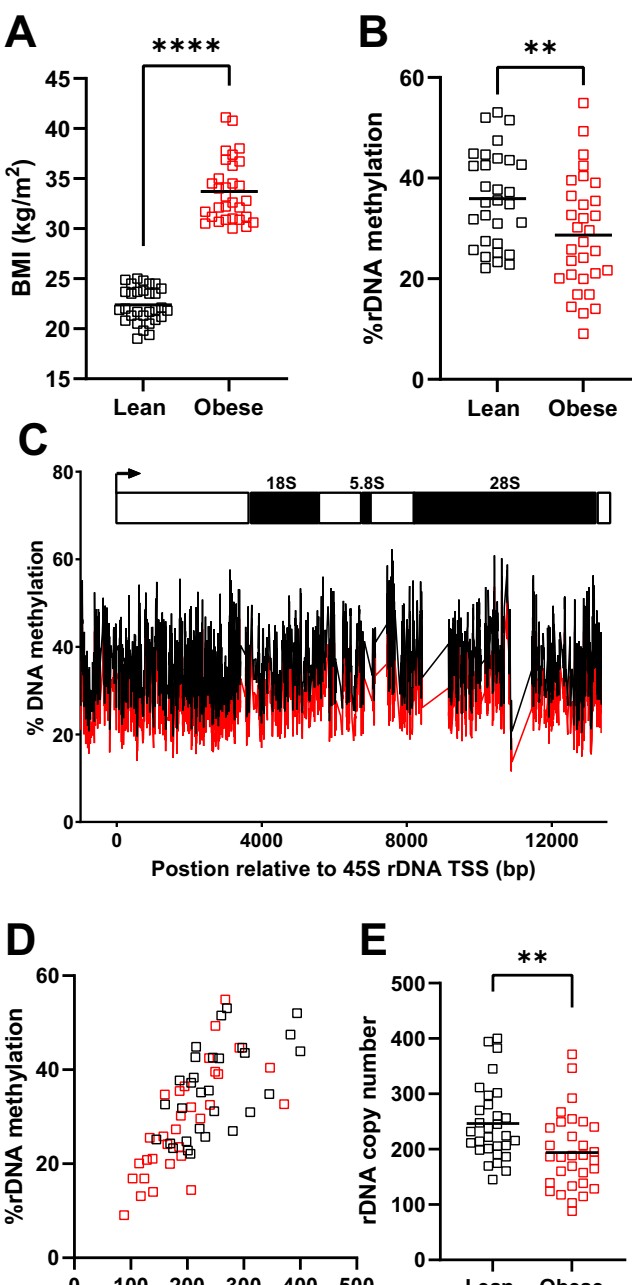

**Fig. 1 | Methylation of rDNA is positively correlated with copy number in blood and associated with obesity. A** The mean body mass index (BMI) of males in an age-matched cohort are different. Lean (BMI < 25 kg/m², n = 31 individual males, black), obese (BMI > 30 kg/m², n = 32 individual males, red), two-sided Mann-Whitney test, p = 9.5 × 10⁻¹². **B** Mean DNA methylation from whole-genome bisulfite sequencing data across the promoter and transcribed region of human 47 S rDNA is lower in obese males. Lean (BMI < 25 kg/m², n = 31 individual males, black), obese (BMI > 30 kg/m², n = 32 individual males, red), two-sided Mann-Whitney test, p = 0.0099. **C** Mean methylation across the transcribed region and 1000 bp upstream of the transcriptional start site is represented for obese (red) and lean (black) males. **D** Mean rDNA methylation across the promoter (−1000 bp upstream) and the entire transcribed region is positively correlated with rDNA copy number (Total cohort: Spearman r = 0.7439, p = 2.2×10⁻¹⁶, n = 63 individual males; Lean only (black): Spearman r = 0.5738, p = 0.0009, n = 31 individual males; obese only (red): Spearman r = 0.7845, p = 9.8×10⁻⁷, n = 32 individual males). **E** Obesity is associated with lower mean total rDNA copy number within this cohort. Lean (BMI < 25 kg/m², n = 31 individual males, black), obese (BMI > 30 kg/m², n = 32 individual males, red), two-sided Mann-Whitney test, p = 0.0024.

direct effects on the production of rRNA transcripts, it must occur earlier during development before epigenetic silencing is fully established. This is supported by the observation that the growth rate in rats from puberty to young adulthood underlies the association with adult body mass.

## Results

### Obese individuals have lower rDNA copy number

Previously, we identified hypermethylated rDNA in inbred adult mice that had been exposed to protein restriction (PR) via maternal diet from the period of conception to weaning[6,7]. This exposure resulted in growth restriction. Intriguingly, the hypermethylation of rDNA was inversely correlated with the weight of the PR-exposed animals at the end of the exposure period. This raised the question of whether the functional genomics of rDNA may be linked to body weight regulation.

To address if altered DNA methylation of repetitive genomic elements, inclusive of the rDNA is associated with body mass index (BMI) in humans, we analysed whole blood from lean (BMI < 25 kg/m²) or obese (BMI > 30 kg/m²) human males (Fig. 1A, p = 9.5 × 10⁻¹²) using whole genome bisulfite sequencing (WGBS). There was no difference in the mean age of the lean and obese groups (~36 years old, Table S1) and no history of diagnosed comorbidities or medication use across the entire cohort. However, there were significant differences between the lean and obese groups relating to anthropomorphic measurements, blood pressure and serum markers (Table S1), consistent with the obese group having a metabolic syndrome associated with higher risk of cardiovascular disease and Type 2 Diabetes[8].

The WGBS data was mapped to a reference genome modified to include a representative copy of the rDNA since the rDNA clusters present on chromosomes 13, 14, 15, 21 and 22 in humans are not included in the genome assembly[5,9]. The sequencing depth of these data (~14x), is sufficient to produce high resolution quantitation of DNA methylation when considering groups of genomic features, or mapping to a consensus for a multi-copy genomic element, such as the rDNA (Table S2). When all reads mapping to genomic features such as exons, introns or a range of repetitive genomic elements were collectively considered, there was no difference in DNA methylation between the lean and obese groups (Fig. S1), with the exception of rDNA, for which methylation was ~7% less across the entire promoter and transcription unit (~14 kB) in the obese compared to lean cohort (Fig. 1B, p = 0.0099 & Fig. 1C). Hypomethylation was also found in the obese group across the key regulatory regions of rDNA, the upstream control element and the core promoter when considered specifically (Fig. S2). This suggests that rDNA is the only genomic feature, when collectively considered, that demonstrates altered DNA methylation in association with obesity.

As we and others have previously shown that DNA methylation levels at rDNA in adult tissues are associated with inter-individual variation in rDNA CN[10–13], we next queried if this epigenetic-genetic interaction is also observed in this cohort. rDNA CN was estimated from the WGBS data using a method based on that for WGS[14]. We and others have previously cross validated this approach[10,13] and also applied strict criteria for data quality control, as this can influence the accuracy of CN assessment[15] (Supplementary Data 1). This was further confirmed by independent cross validation using digital droplet PCR (ddPCR) to estimate copy number using a sequencing-independent methodological approach (Fig. S3). Consistent with previous reports, we observed that rDNA CN and DNA methylation are highly correlated in blood, such that individuals with higher rDNA CN also have more rDNA methylation (Fig. 1D, Spearman *r* = 0.74, *p* = 2.2 × 10⁻¹⁶). This relationship was present in both the lean (Spearman r = 0.5738, p = 0.0009) and obese (Spearman r = 0.7845, 9.8 × 10⁻⁷) groups when considered separately as well as overall. Reconcilable with the observation of hypomethylation of rDNA in the obese group, rDNA CN was also significantly lower in the obese compared to lean group (Fig. 1E,

p = 0.0024) both when calculated from WGBS and using ddPCR (Fig. S3). This was not accounted for by differences in the ethnic composition of the lean and obese groups (Table S3), or sequencing parameters (Fig. S4). Furthermore, the trend for lower CN in obese individuals was observed in multiple ethnicities (Fig. S5). These observations demonstrate that total rDNA methylation and CN are positively correlated in humans and that rDNA CN variation in blood is associated with variation in a complex trait in humans, adult BMI.

## The strength of association is influenced by interventions

To validate the observed association between rDNA CN and BMI, we re-analysed published reduced representation bisulfite sequencing data (RRBS) derived from the adipose tissue of Finnish males 45–67 years of age from the METSIM study[16,17]. We devised a methodology for estimating rDNA CN from RRBS for determining relative (rather than absolute) CN across individuals. The method was cross-validated with the previously published method used for the WGBS data above, which in turn we have previously validated using an independent methodology, ddPCR (and Fig. S6 and Fig. S3). In this cohort, we also observed a strong positive correlation between rDNA CN and methylation, confirming its presence in at least two different human tissues (Fig. S7).

The individuals in the METSIM cohort were not selected for clinical obesity (BMI > 30 kg/m$^2$). Therefore, a continuous spectrum of BMI is represented with the majority of individuals included in this analysis classified as clinically "overweight" (25 < BMI < 30 kg/m$^2$). As such, we performed a correlation analysis between the relative rDNA CN and an individual's BMI, again revealing a negative correlation that passed a nominal significance threshold (Fig. 2A. r = −0.18, p = 0.02). Consistent with this observation, rDNA methylation also demonstrated a negative correlation with BMI, although the effect did not pass the nominal threshold of p < 0.05 (Fig. 2B, r = −0.13, p = 0.09). As this cohort is older and has a larger age range than the discovery cohort, we calculated age-adjusted BMI and found that this did not have a significant effect on the correlation between rDNA CN or methylation with BMI (Fig. S8).

The METSIM validation cohort has a significant proportion (69/169) of individuals regularly taking one or more medications, most commonly statins. Therefore, we examined the association of medication status with anthropomorphic and metabolic measurements. Although there was no difference in age or BMI in the medicated compared to the non-medicated groups, the medicated group had elevated measurements for waist circumference, waist to hip ratio, diastolic blood pressure, fasting insulin levels and the Homeostatic Model Assessment for Insulin Resistance (HOMA-IR) (Table S4). Collectively, these findings indicate a higher prevalence of metabolic syndrome in these individuals[8]. However, unlike the younger discovery cohort, this was not reflected in higher low-density lipoprotein levels, consistent with mitigation by statin use[18].

To investigate whether medication or other interventions that modify anthropomorphic and metabolic phenotypes might influence the strength of genotype-phenotype associations, we next considered the medicated and non-medicated groups separately. Intriguingly, the correlation between BMI and both rDNA CN and methylation strengthened when the non-medicated group was considered separately (Fig. 2C, r = −0.30, p = 0.0025 & Fig. 2D, r = −0.25, p = 0.01), but disappeared when the medicated group were analysed alone (Fig. 2E, r = 0.05, p = 0.71 & Fig. 2F, r = 0.07, p = 0.59). This could not be explained by differences in sequencing quality (Fig. S9) after sample exclusion based on strict QC of the sequencing data (Supplementary Data 2). Similar findings were observed after adjusting BMI for age (Fig. S10). Taken together, these results support a quantitative relationship between rDNA CN and BMI in adults. This has been observed in two independent, ethnically different populations and holds across two different tissues. However, we find that lifestyle interventions that influence phenotype can reduce the genotype-phenotype correlation.

Comparison of the relationship between rDNA CN and the anthropomorphic and metabolic traits captured in both cohorts suggests that body mass is the primary measured trait associated with rDNA CN variation, as BMI and waist circumference were the only variables significantly associated across both the extreme discovery and METSIM validation cohorts (Table S5). Other variables did show cohort-specific negative correlations with rDNA CN, such as C-reactive protein, fasting insulin levels and the HOMA-IR exclusively in the discovery cohort and diastolic and systolic blood pressure in the unmedicated subgroup of the replication cohort. As these are all established sequelae of being overweight, it is likely this is explained by differences in the age and phenotypic extremes between the cohorts. rDNA methylation was less strongly correlated with BMI and other traits than rDNA CN, supporting the conclusion that it is the genetic rDNA CN association with BMI that is driving the altered DNA methylation profiles we initially observed.

## rDNA CN does not systematically vary by cell-type

Ageing[19] and body mass[20] have been associated with changes in blood cell proportions and adipose infiltration[21]. However, total rDNA CN has previously been shown to be consistent across multiple tissues in mice[22] and more recently humans[13]. We further confirmed this specifically by looking at variation in purified cell populations, rather than whole tissues to investigate whether cell-type specific variation in rDNA CN is present in multiple purified blood and other cell types from donors[23] (Fig. S11). This confirmed that although there is extensive inter-individual rDNA CN variation, there is no systematic rDNA CN difference across cell types. Therefore, the rDNA CN association with BMI is not a downstream artifact of altered blood cell composition across samples.

## rDNA CN is not discordant in twins with divergent BMI

The observation that rDNA CN negatively correlates with adult BMI raises questions about the origin of the association. Two scenarios are plausible, i) germline inherited rDNA CN, through an unknown mechanism, can influence BMI variation, or ii) environmental exposures and/or metabolic changes occurring concomitantly with increasing BMI may lead to rDNA (epi)genetic instability and rDNA CN loss. To our knowledge, there is no precedent for a human trait associated with germline inherited rDNA CN, but rDNA CN changes have been observed in human cancers[24,25].

To address the issue of causation, we obtained RRBS data derived from whole blood of monozygotic twins discordant for BMI and of a single ethnicity. The original study excluded individuals with diagnosed comorbidities[26]. The cohort characteristics for the 10 male and 14 female monozygotic twin pairs included in this analysis after data quality control (Supplementary Data 3, Fig. S12) are shown in Table S6. The twins were confirmed to have highly discordant BMI (Fig. 3A, p = 1.9 × 10$^{-5}$), with both the leaner and heavier twins spanning all clinical BMI group classifications. However, leaner co-twins were enriched in the clinically "lean" range (BMI < 25 kg/m$^2$), and the heavier co-twins enriched in the clinically "overweight" range (25 kg/m$^2$ < BMI < 30 kg/m$^2$). Despite discordance in BMI, there was no difference observed for the rDNA CN within twin pairs (Fig. 3B, p = 0.84). Consistent with this, there was not any association between within-twin rDNA CN variation and twin age, or BMI discordance (Fig. S13). As with the previous cohorts, there was a positive correlation between rDNA CN and methylation (Fig. S14), and in line with this, there was no rDNA methylation differences between discordant co-twins (Fig. 3C, p = 0.29). These results, together with the cell type specific analysis above, support the idea that rDNA CN is not subject to extensive random drift over time or in response to environmental or metabolic changes associated with altered BMI. It is therefore more

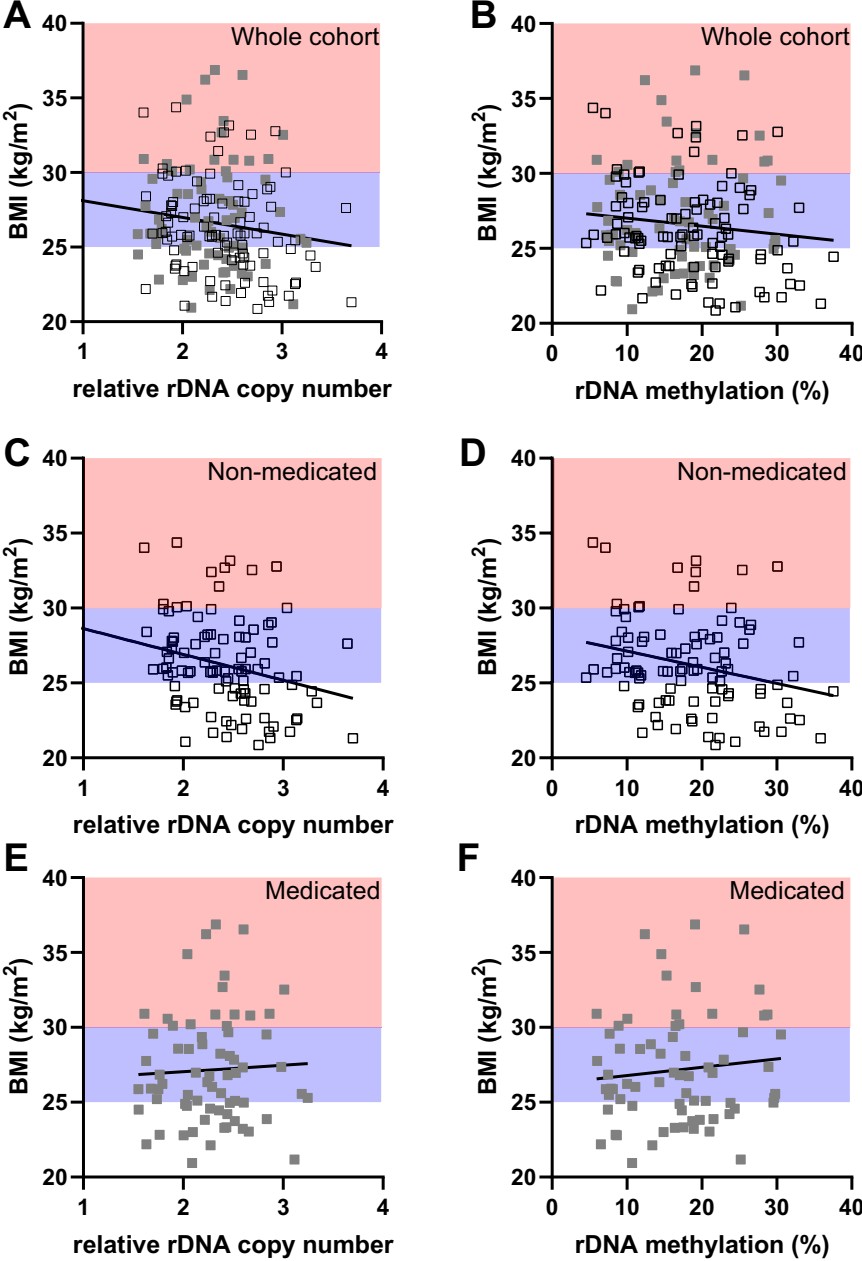

**Fig. 2 | rDNA copy number negatively correlates with BMI but the association is diminished by lifestyle interventions.** BMI clinical classifications are indicated (lean (BMI < 25 kg/m²) white background, overweight (25 kg/m² < BMI < 30 kg/m²) blue background, obese (BMI > 30 kg/m²) red background). **A** Relative rDNA copy number is negatively correlated with BMI when the entire cohort is considered (Two-sided Spearman r = −0.1785, p = 0.0202, n = 169 individual males, non-medicated (open black squares), medicated (full grey squares)). **B** rDNA methylation is not correlated with BMI when the entire cohort is considered (Two-sided Spearman r = −0.1320, p = 0.0872, n = 169 individual males), non-medicated (open black squares), medicated (full grey squares)). **C** Relative rDNA copy number is negatively correlated with BMI when only non-medicated individuals are considered (Two-sided Spearman r = −0.2992, p = 0.0025, n = 100 individual males (open black squares)). **D** rDNA methylation is negatively correlated with BMI when only non-medicated individuals are considered (Two-sided Spearman r = −0.2518, p = 0.0115, n = 100 individual males (open black squares)). **E** Relative rDNA copy number is not correlated with BMI when only medicated individuals are considered (Two-sided Spearman r = 0.04607, p = 0.7070, n = 69 individual males, full grey squares). **F** rDNA methylation is not correlated with BMI when only medicated individuals are considered (Two-sided Spearman r = 0.06675, p = 0.5858, n = 69 individual males, full grey squares).

likely that it is germline inherited rDNA CN that is associated with adult BMI.

### rDNA CN is not associated with other known genetic variation

As BMI has a strong genetic as well as environmental component, we next sought to address whether there is a direct association between rDNA CN and previously identified single nucleotide variants (SNVs) associated with BMI from GWAS. To this end, we retrieved summary statistics from a meta-analysis of BMI including ~700,000 individuals[27] and utilised these as base data for calculating a BMI polygenic risk score (PRS) for individuals in the 1000 Genome Project[28]. Only individuals which have also independently been determined to have WGS data of sufficient quality for rDNA CN estimation were included[15]. PRS scores calculated using only the highly significant, near-independent SNVs previously identified[27], or a more relaxed significance threshold produced highly correlated PRS scores, as expected (Fig. S15). We then asked if the PRS calculated from BMI-associated SNVs explains variance in rDNA CN. We

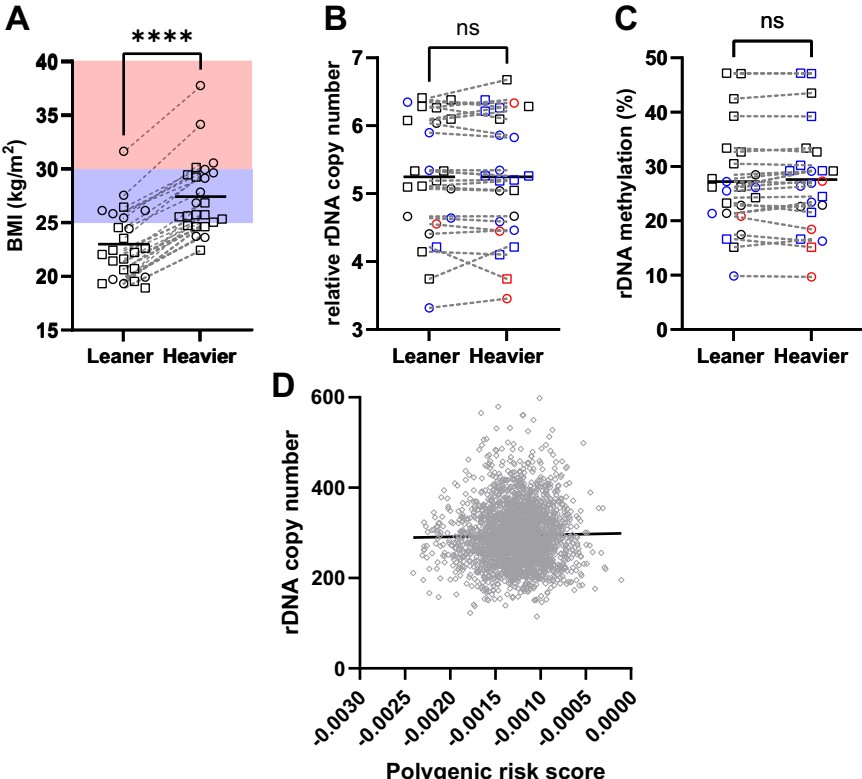

**Fig. 3 | BMI does not induce rDNA copy number variation and is not influenced by BMI-associated genetic variation in the rest of the genome. A** Mean BMI is discordant between monozygotic twins (Two-sided Wilcoxon matched-pairs signed rank test, $p = 1.9 \times 10^{-5}$, n = 24 twin pairs) and significantly paired (Two-sided Spearman r = 0.8723, $p = 2.8 \times 10^{-8}$, n = 24 twin pairs). BMI clinical classifications are indicated (lean (BMI < 25 kg/m²) clear background, overweight (25 kg/m² < BMI < 30 kg/m²) blue background, obese (BMI > 30 kg/m²) red background. Sex is indicated by symbol shape, circle = female (10 MZ twin pairs), square = male (14 MZ twin pairs). Each symbol represents an individual. **B** Mean relative rDNA copy number is not different between twins (Two-sided Wilcoxon matched-pairs signed rank test, p = 0.8388, n = 24 twin pairs) and is significantly paired (Two-sided Spearman r = 0.9817, $p = 1.5 \times 10^{-6}$, n = 24 twin pairs). BMI clinical classifications are indicated (lean (BMI < 25 kg/m²) black, overweight (25 kg/m² < BMI < 30 kg/m²) blue, obese (BMI > 30 kg/m²) red. Sex is indicated by symbol shape, circle = female (10 MZ twin pairs), square = male (14 MZ twin pairs). Each symbol represents an individual. **C** Mean rDNA methylation is not different between twins (Two-sided Wilcoxon matched-pairs signed rank test, p = 0.2929, n = 24 twin pairs) and is significantly paired (Two-sided Spearman r = 0.9704, $p = 1.2 \times 10^{-6}$). BMI clinical classifications are indicated (lean (BMI < 25 kg/m²) black, overweight (25 kg/m² < BMI < 30 kg/m²) blue, obese (BMI > 30 kg/m²) red. Sex is indicated by symbol shape, circle = female (10 MZ twin pairs), square = male (14 MZ twin pairs). Each symbol represents an individual. **D** BMI polygenic risk scores are not correlated with rDNA copy number (Two-sided Spearman r = 0.01284, p = 0.5304, n = 2390). Each symbol represents an individual human.

did not find any models that could explain variance in rDNA CN (top model ($p < 0.005$ threshold) goodness of fit $R^2 = 0.00063$, p = 0.19). This is reflected when the PRS scores are directly correlated with the rDNA CN estimates (Fig. 3D, r = 0.01284, p = 0.5304). In an alternative approach, we also performed a GWAS for rDNA CN in the same cohort. Only one SNV was identified that passed the nominal $p < 5 \times 10^{-8}$ threshold for genome-wide correction (Chr5: 159614102, $p = 3.7 \times 10^{-8}$). This position is only variable in some East Asian populations[29], suggesting that it is unlikely to be confounding results in the predominantly European discovery and validation cohorts. Furthermore, this SNV has not previously been associated with body mass or related traits. Taken together, these findings suggest that rDNA CN is not strongly influenced by common inter-individual genetic differences in other parts of the genome that contribute to BMI variation.

## rDNA CN is associated with post-pubertal growth rate in rats

We next asked whether an association between rDNA CN and body mass also exists in non-human mammals. To address this, we leveraged published RRBS data generated from the liver of female, outbred (Sprague-Dawley) rats[30,31]. The authors made available weight data that was longitudinally collected throughout the experimental period, weeks 8-19 of age. This period encompasses the onset of sexual

maturity in this strain (9-10 weeks of age for females) and is still within a period of growth, which has been reported to extend to 24 weeks of age[32].

After excluding some individuals based on strict quality control of the RRBS data (Fig. S16 and Supplementary Data 4), we further verified that there was no effect of the study treatments on either weight, rDNA CN or methylation (Fig. S17). Plotting the longitudinal weight data confirmed that this was still a period of active growth (Fig. 4A). The availability of longitudinally collected data allowed us to query whether growth rate correlates with rDNA CN. Interestingly, cross-sectional analysis of each time point demonstrated that there was no correlation at early timepoints (Fig. 4B, r = −0.072, p = 0.6432), with a negative correlation only emerging towards the end of the experimental period, reaching significance at weeks 18 and 19 (Fig. 4C, r = −0.35, p = 0.0206 & Table S7). Indeed, this could be explained by a negative correlation between the weight gained over the study period and rDNA CN (Fig. 4D, r = −0.41, p = 0.0055), but not methylation (Fig. S18), despite a positive correlation being observed between rDNA CN and methylation in the tissues at harvest, as in all other analysed datasets (Fig. S19). Collectively, these findings suggest that the correlation between rDNA CN and body mass is not unique to humans and furthermore, becomes manifest between the ages of puberty and early adulthood.

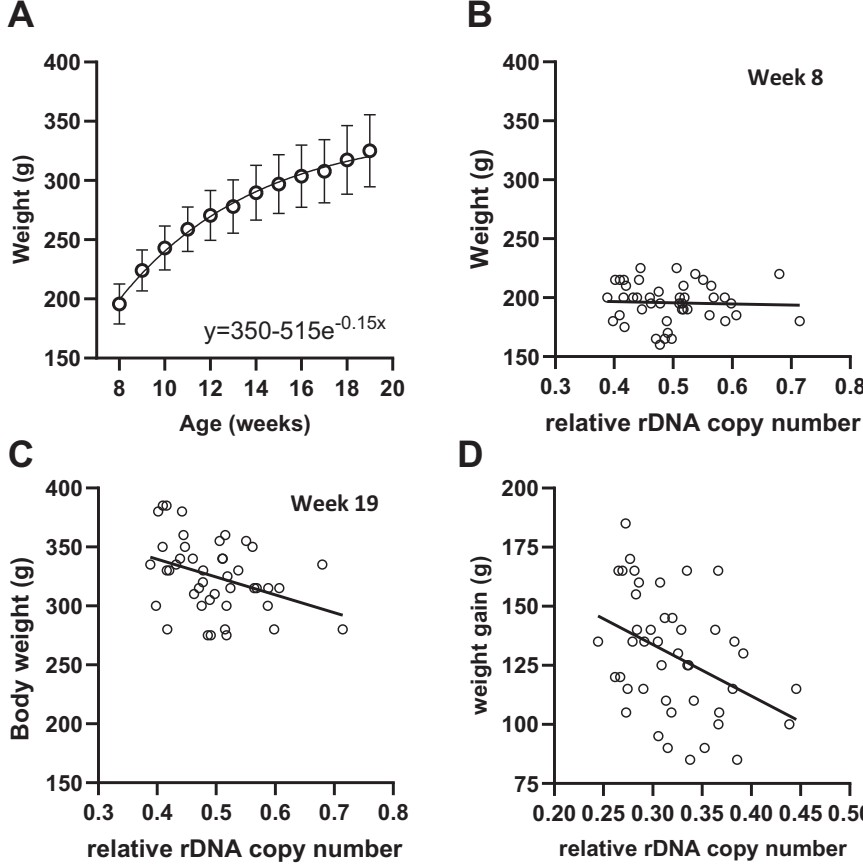

**Fig. 4 | Weight gain from puberty to early adulthood negatively correlates with rDNA copy number in Sprague-Dawley rats.** RRBS data derived from liver of female Sprague-Dawley rats at the time of sacrifice. **A** Weekly weight data fits an exponential plateau curve ($R^2 = 0.7232$, mean and standard deviation represented). **B** Absolute weight is not negatively correlated with rDNA copy number at week 8 (Two-sided Spearman $r = -0.07181$, $p = 0.6432$, $n = 44$ individual rats). **C** Absolute weight is negatively correlated with rDNA copy number at week 19 (Two-sided Spearman $r = -0.3481$, $p = 0.0206$, $n = 44$ individual rats). **D** Weight gain (week 8–19) is negatively correlated with rDNA copy number (Two-sided Spearman $r = -0.3977$, $p = 0.0075$, $n = 44$ individual rats).

## rDNA CN is not correlated with rRNA transcription in adult tissues

Having demonstrated an association with rDNA CN and body mass regulation across two mammalian species, we next asked whether rDNA CN variation is associated with nascent rRNA transcript levels in adult somatic tissues. Previously, we have shown that expression of specific, well-defined rRNA haplotypes in mice reflects their relative contribution to total rDNA CN after adjusting for the silencing of the methylated copies[10]. Similar haplotypes in humans are yet to be defined. Comparing the total rDNA copy number before and after adjusting for methylation in three independent data sets derived from inbred mice, outbred mice and human lymphoblastoid cell lines (LCLs) did not reveal an association with nascent rDNA transcription rates (Fig. S20). This supports the conclusion that the association between rDNA CN and body mass is not due to altered transcription rate in adult tissues and if the mechanism is associated directly with rDNA transcription, then this occurs earlier in development, prior to compensation of rDNA CN variation through epigenetic silencing upon the initiation of differentiation[33]. This finding agrees with the timing at which the emergence of the genotype-phenotype association was observed in the rats.

## Discussion

rDNA CN is highly variable in humans[9,10,15]. This variation is thought to derive from a very high frequency of meiotic rearrangement at the rDNA repeats[34]. However, germline genetic variation at rDNA has not, to the best of our knowledge, been studied in relation to human trait variation as it requires sequencing data rather than genotyping arrays. Here we provide evidence that germ-line variation in rDNA CN is associated with body mass in mammals using multiple cohorts and tissues, and both extreme and continuous phenotypes.

Somatic rDNA CN instability has been reported previously in human diseases, such as cancer and neurodegenerative disorders[22,24,25,35]. However, in these cases there is clear evidence that rDNA CN variation is consequential of the disease process and shows no directional association, or predictive value. There are some rare examples of cancers with specific genetic lesions that do lead to tumour-specific rDNA CN loss[22,36]. These are restricted to epigenetic regulators important for the maintenance of heterochromatin at repetitive genomic regions, including the rDNA[36] or tumour suppressors involved in chromatin stability[37]. However, the evidence from the monozygotic twins and multiple cell types from the same donors analysed here suggests that rDNA CN does not undergo a high degree of random genetic drift, suggesting somatic stability.

A caveat of our findings is that at this stage we are limited to identifying an association between rDNA CN and body mass. Changes in inflammatory cell types are known to occur in obesity[38]. However, our analyses demonstrate that rDNA CN does not systematically vary with cell type and furthermore, the association of rDNA CN with growth rate in rats in the absence of an obesogenic environment suggests that the origin of the association is not with obesity per se, but rather through a mechanism that more fundamentally regulates organismal growth. Interestingly, a human longitudinal study has recently linked the rate of BMI change during the post-pubertal to

young adult ages to higher risk for obesity in mid-life, independently of the actual BMI during this period[39]. This implies that factors influencing growth rate during this period are associated with later disease risk.

Our data suggests that epigenetic silencing, associated with DNA methylation compensates for higher rDNA CN, essentially serving to "normalise" the rate of rRNA transcription in adult tissues. This, together with the phenotype-genotype association emerging by early adulthood, suggests that the physiological basis for the association between rDNA CN and BMI may occur earlier in development. The dynamic regulation of rDNA transcription is essential for embryonic development in mammals[33], with pluripotency requiring a lack of rDNA silencing to produce a translational state essential for maintaining stemness and inhibiting genes required for differentiation[37,40]. Silencing of the rDNA units begins with cellular differentiation but stabilises slowly. Could then rDNA CN influence cell fate decisions at these earlier timepoints? Interestingly, disease caused by mutations in ribosomal proteins producing a reduction in total ribosome levels, without a change in composition have been shown to influence cell fate decisions through selectively influencing translation of lineage specifiers[41]. This raises the intriguing possibility that variation in rDNA CN may produce subtle effects on cell commitment in development which ultimately impact growth trajectories later on. An alternative hypothesis is based on observations from Drosophila, where variations in 35S-rDNA CN have been shown to influence the expression of other genes by altering genome-wide chromatin structure[42,43]. Exploring these hypotheses is beyond the scope of the work presented here and will be challenging, as rDNA CN in mammals is not amenable to manipulation by reverse genetic approaches, even in the era of CRISPR. This is due to the large tandem arrays of rDNA with very high sequence homology, combined with a lack of knowledge of their organisation beyond the recently released telomere-to-telomere genome assembly which is derived from a single cell line from a hydatiform mole that is homozygous[5]. Nonetheless, the first demonstration that rDNA CN may be associated with a biomedically relevant human phenotype provides the impetus for such investigations in the future.

## Methods

### Sample information

**Mixed ethnicity obese discovery cohort.** Participants were recruited as part of a prospective cohort study, the Dad's Health Study at University College Hospital London (UCLH) to investigate the association between paternal metabolic health (including lean and obese men) and offspring birth weight, May 2016−March 2019. Whole blood samples were collected from participants. Participants were phenotyped with regards to BMI, waist circumference, systolic and diastolic blood pressure, blood lipids, fasting insulin and glucose levels and C-reactive protein (CRP). Two groups of participants were included; lean (BMI < 25 kg/m$^2$) and obese (BMI > 30 kg/m$^2$). Summary phenotypic data for each group is detailed in Table S1. BMI was determined with light clothing, by a trained researcher in the same visit at which blood samples were obtained. Peripheral blood samples were centrifuged at 3000 g for 15 minutes within one hour of venepuncture and the buffy coat stored at -80$^0$C. Only male participants were recruited based on self-reported sex and verified by reads mapping to the Y chromosome from the WGBS data.

**Ethics approval and consent to participate.** Ethical approval for the study was granted from the South East Coast−Surrey Research Ethics Committee on 28 September 2015 (REC reference number 15/LO/1437, IRAS project ID 164459). The study was also registered with the University College London Hospital Joint Research Office (Project ID 15/0548). All participants provided written, informed consent.

All collaborating authors have been acknowledged in accordance with inclusion and ethics relevant to global research.

## Data generation

**DNA extraction.** DNA was extracted from 200 ml of buffy coat using the Qiagen QIAamp DNA Blood Mini Kit (Qiagen, Cat No. 51106) according to the manufacturer's instructions and including RNA digestion. Purity of the extracted DNA was confirmed on a Nanodrop (ThermoFisher, cat. No ND-ONEC-W) and the concentration determined using the QuBit dsDNA HS Assay Kit (ThermoFisher, Cat No. Q32854).

**Whole genome bisulfite sequencing library construction and sequencing.** Genomic DNA was diluted to 10 ng/µl, and 100 µl sonicated using a Bioruptor® Pico (Diagenode, Cat No. B01060010) to achieve a 500-600 bp size range, which was confirmed using a TapeStation High Sensitivity D1000 System (Agilent, Cat No. 5067-5584 & 5067-5585). Once the desired size range was achieved, 200 ng of sonicated DNA was subjected to bisulfite conversion using the EZ DNA Methylation-Gold™ Kit (Zymo, Cat No. D5006). Libraries were then made using the Accel-NGS Methyl-Seq DNA Library Kit with unique dual indices following the size-selection guidance provided with the kit and 10 cycles of amplification to minimise clonality. The removal of all adaptors and quantification were confirmed with TapeStation and QuBit before libraries were pooled into equimolar 12-plex pools and subjected to 150 bp paired-end sequencing on a NovaSeq6000 (GeneWiz).

**Human digital droplet PCR for rDNA copy number.** 1 µg of human genomic DNA was digested with NsiI in rCutSmart buffer (NEB, Cat No. R3127) for 1 hour at 37 °C, followed by heat inactivation. Digests were cleaned up using the DNA Clean and Concentrator-5 kit (Zymo, Cat No. D4014) using a 5:1 binding buffer to sample volume:volume ratio. Eluted DNA concentrations were determined using the QuBit dsDNA High sensitivity kit (ThermoFisher, Cat No. Q32854), diluted to ~0.5 ng/µl in nuclease-free water and then verified using the QuBit assay once more. This concentration was determined empirically using a standard curve to optimise the conditions for ddPCR. ddPCR reactions consisted of 1x Absolute Q$^{TM}$ DNA Digital PCR Master Mix (Thermofisher, Cat No. A52490), ~0.5 ng of NsiI digested genomic DNA, 1x TaqMan RNase P-Vic (Thermofisher, Cat No. A30064) and 1x Taqman 18S-FAM (Thermofisher, custom assay design) in a final volume of 10 µl. The custom primer/probe combination was designed using Genbank accession KY962518.1, the forward primer sequence was (5′-CCGCGGTTCTATTTTGTTGG-3′), the reverse primer sequence was (5′-CTGATCGTCTTCGAACCTCC-3′) and the probe sequence was (5′-CGAATGCCCCCGGCCGTCCC-3′). RNase P-VIC was used as a validated single copy gene reference. Thermal cycling conditions were 10 min at 96 °C, followed by 40 cycles of 5 s at 96 °C then 15 s at 60 °C on the QuantStudio Absolute Q Digital PCR System. Relative copy number using the QuantStudio Absolute Q Digital PCR Software (v6.3.0), with the CNV set to the FAM channel and CNV-REF set to the VIC channel with a value of 2. Samples were excluded is the Lambda (Cp/Rxn) values were ≥1.6. CN estimates and associated Lambda values are reported in Supplementary Data 5.

**Mouse digital droplet PCR for rDNA copy number.** This assay was performed as described previously[10].

**Mouse nascent 47S-rRNA qRTPCR.** This assay was performed on kidney samples from mice of different strains as described previously[7].

**Human nascent 47S-rRNA qRTPCR.** RNA was isolated using TRIzol as per manufacturer's instructions, quality assured on a RNA 6000 Nano Chip (Agilent) and 500 ng of total RNA was reverse transcribed using random hexamers (NEB ProtoScript®II). Real-time qPCR was performed using QuantiTect SYBR Green qPCR mix (Qiagen). Primers to amplify the precursor of the human rRNA and the housekeeping

control ACTB were taken from previously published work and GAPDH primers designed as F- 5′-CCATCACCATGTTCCAGGAG-3′, and R- 5′-CCTGCTTCACCACCTTCTTG-3′[44–46].

## External data sources

**Justification for cohort selection.** rDNA is only captured using long or short-read sequencing approaches. Therefore, for methylation analysis, we were restricted to selecting cohorts analysed using bisulfite-sequencing based approaches. Furthermore, the strong positive correlation between rDNA CN and methylation provided a useful additional quality control. This additional quality control is absent in WGS datasets. Therefore, we limited our analysis of WGS data sources for which the rDNA CN had been previously established and rigorously quality assessed[15].

**Validation cohort (METSIM adipose tissue).** Raw RRBS data for this cohort was downloaded from GEO (GSE87893). Limited phenotype data for the samples included were made available through collaboration[16]. This pre-existing data resource consisted solely of participants of male sex, as previously described[16]. Summary phenotypic data from this cohort for the samples included in these analyses can be found in Table S4.

**Monozygotic twin cohort.** Raw reduced representation bisulfite sequencing data from the monozygotic twin cohort was made available through collaboration and similarly, can be made available by request to Q. Tan[26]. Data is derived from whole blood from twins that have no diagnosed illness or medications. Summary phenotypic data from this cohort for the twin pairs included in this analysis are provided in Table S6. Both male and female twin pairs were included and sex verified by mapping reads to the sex chromosomes[26]. Sex was not included in the analyses as they were all paired across the twins.

**1000 genome project.** rDNA CN estimates and single nucleotide variant calls (SNV) were obtained from published sources[15,46]. Both sexes are included in this analysis and sex is included as a cofactor in analyses.

**Rat data with longitudinal weights.** Raw RRBS data for this cohort was downloaded from GEO (GSE157551). Longitudinal weight measurements were made available through collaboration[30,31]. The original study included only female rats.

**Methylation atlas.** WGBS data generated from sorted and pure human cell populations was retrieved from the European Genome Archive (Dataset ID EGAD00001009789). Only individuals with multiple cell types were included in the analyses, with both sexes included and determined as specified in the original publication. Sex was not included in the analyses. This left 67 samples from 18 donors and included 26 cell types represented by 2 or more donors.

**Mouse data for rDNA copy number and/or methylation quantitation.** This data if not described above has been generated as part of previously published work[7,10]. Data from previous work was from exclusively male mice.

## Data analysis

**Reference sequences.** Genomic reference sequences used throughout this study were generated as follows to alleviate potential coverage loss and spurious alignments. The repetitive element appearing in the rDNA IGS closest to the 3′ end of the unit was identified from the publicly available annotations for the human rDNA unit reference (Genbank accession KY962518.1). The midpoint of this repetitive element, which is located 2120 base pairs upstream of the TSS, was employed as breakpoint for creating a "looped" rDNA unit. In

particular, the bases downstream of the breakpoint up to the end of the rDNA unit reference were prepended to the bases upstream of the breakpoint, which improves read coverage around the TSS by avoiding reads being discarded due to split alignments.

To minimise the risk of sequencing reads from locations outside the rDNA being spuriously mapped to the rDNA, identified rDNA pseudocopies in the Hg38 assembly were masked, and the "looped" rDNA reference mentioned above was appended. Masked regions and their genomic coordinates, including an entire rDNA unit located in an unplaced contig (Genbank accession GL000220.1) are shown in Table S8.

A human exome reference was obtained and adapted as described previously[14], with exon sequences and their annotations being downloaded from the EMBL/EBI repository. In particular, exons from the sex chromosomes and smaller than 300 bases were removed, as were sequences with significant similarity as reported by blastn version 2.7.1+ in --ungapped mode. This left a total of 12,898 exon sequences in the adapted reference.

In the case of rat data, the most complete rDNA consensus sequence available presently spans solely from the 5′ end of the 18 S to the 3′ end of the 28 S (Genbank accession V01270.1). A blastn comparison between V01270.1 and the rat whole genome assembly Rn7 revealed several apparent partial rDNA pseudocopies scattered across the assembly, plus three end to end matches. These three complete pseudocopies, whose coordinates are indicated in Table S8, were thus masked. To minimise the potential detrimental effects of the partial pseudocopies, only the sequence corresponding to the 18 S (positions 1 to 1874) was appended to the masked Rn7 assembly as an additional contig.

**Sequencing data processing.** Initial processing of data was performed using fastqc version 0.11.9 to identify failed libraries for exclusion. Data was then trimmed for base quality and adaptor removal using trimgalore version 0.6.5, with the --paired and --rrbs options enabled when appropriate. For WGBS data, parameters --clip_r1 10 and --clip_r2 20 were also enabled to improve further alignment. Remaining parameters were set to the default values.

Deduplication was not performed on WGBS. The repetitive nature of the rDNA within the genome increases the probability of excluding the significant number of associated reads in addition to PCR artifacts from further analysis. Deduplication should not be performed on RRBS as recommended by Bismark processing protocol.

Alignments to the reference sequence were performed using bismark version 0.23.0 with underlying bowtie2 for the WGBS and RRBS data sets. Bisulfite conversion of the reference sequence for WGBS/RRBS alignments was performed using bismark_genome_preparation. Alignment output files were then sorted, indexed and filtered to retain only reads uniquely mapping to the rDNA reference using samtools version 1.10.

Methylation data from the WGBS/RRBS data was extracted from the bismark alignments using bismark_methylation_extractor.

**Data QC and sample exclusion.** Samples were excluded on the basis of bismark report data (Tables S3, S6, S8, S10). Namely, if they had extremely high or low uniquely mapped reads, poor mapping efficiency, poor bisulfite conversion (as evidenced by high non-CpG methylation values). In the case of the RRBS datasets, samples that were extreme outliers with regards to total CpG methylation were also excluded as this indicates a potential problem with enzyme digestion or size selection altering the genomic regions captured compared to other samples in the same set.

**rDNA CN estimation.** rDNA CN was estimated from WGBS using a previously described method[10,14]. This involved aligning reads to the exome reference described above to obtain the average read depth for

each sample using samtools depth. The average read depth mapped to the 18 S subunit from the whole genome + rDNA alignments were then normalised to the exome value and CN calculated as 2 × (18 S /exome average read depth) for each sample. We previously validated this method of estimating the relative total CN of rDNA across samples by comparison to digital droplet PCR[10] as well as here (Fig. S4).

To estimate the relative rDNA CN across samples from the RRBS data, we developed an alternative approach due to the patchy coverage of exomes in these libraries. In this approach the number of reads aligned to the rDNA reference as reported by samtools idxstats was divided by the total number of alignments listed in the corresponding bismark report. This method was cross-validated with the methodology using whole-genome data and showed a high correlation (Fig. S4).

**QC of rDNA CN estimation.** The effect of depth coverage on rDNA CN estimation was analysed using library down-sampling as described elsewhere[13]. Four datasets of high comparable coverages were selected across WGBS (D284 and D454) and RRBS (SRR4418992 and SRR4418945) data. After trimming was performed, the trimmed reads were split into 10 subsamples of equal size using the seqkit version 2.6.1. Subsamples then were randomly selected and merged to achieve a coverage of 90%, 80%, 70%, etc. rDNA CN were estimated from merged subsamples as previously described. This confirmed that small variations within our data parameters were not influencing CN estimation in our analyses (Fig. S21).

**Genomic feature methylation and coverage estimates.** Genomic feature methylation was extracted using R package methylKit version 1.16.0[47]. Genomic features and repetitive DNA elements were defined based on hg38 genomic annotation Reference Sequence (RefSeq) and RepeatMasker database acquired from UCSC table browser respectively[48]. Regions 1 kb upstream and downstream of the transcription start site of the reference genome were considered as promoters. For rDNA analysis, only CpGs covered by at least 50 unique reads were used whereas all CpGs were analysed for rest of the genome. Coverages per genomic features were estimated by calculating average number of reads that cover all CpG sites within each annotated genomic feature for each sample revealed by methylKit methRead() function.

**Genotype association between BMI PRSs and rDNA CN.** The base data were obtained from the meta-analysis in ref. 27, with files hosted within the GIANT consortium data site of the Broad Institute. In particular, the two summary files for BMI analysis (updated after June 25, 2018) were retrieved, one containing all considered loci (hereinafter referred to as the "COMPLETE" file, with 2,336,269 variants), and another including Conditional and Joint (COJO) -transformed p-values of only significant hits (hereinafter, "COJO" file, with 941 variants).

Base data SNPs were filtered for quality using the munge_sumstats.py script from ldsc version 1.0.1[49], with the options --snp SNP, --N-col N, --a1 Tested_Allele, --a2 Other_Allele, --frq Freq_Tested_Allele_in_HRS and --n-min 100000 for both COMPLETE and COJO input files. For the COJO input, the options --p P_COJO, --signed-sumstats BETA_COJO,0 and --ignore P,SE,BETA were also included to ensure the intended estimates were employed. This filtering left 1,977,697/2,336,269 SNPs from the COMPLETE input and 802/941 from the COJO input.

Although not explicitly specified, the genomic coordinates indicated in the base data files appear to refer to the GRCh37 assembly (e.g., rs1000096 is listed at chr4:38,692,835, instead of chr4:38,691,214 as it would correspond in GRCh38). For consistency with the target data, remaining locations in the base data were thus converted to GRCh38 coordinates using the liftOver function of the rtracklayer package version 1.54.0 in R with the hg19toHg38.over.chain file obtained from UCSC.

Target data vcf files were retrieved from the 1000 Genomes Project FTP servers. In particular, information for 3202 individuals was obtained for the autosomal chromosomes from the 20201028_3202_raw_GT_with_annot folder within the 1000G_2504_high_coverage collection. These were initially filtered at chromosome level with plink2 (version 2.0-20200328) --make-bed, keeping solely the SNPs remaining on the base data using the --extract bed1 option, and QC parameters --mind 0.01, --geno 0.01, --maf 0.01, --hw2 1e-6, and --max-alleles 2. The --set-missing-var-ids @:# option was also included to avoid apparent duplicate names, and a list of successful loci was requested with the --write-snplist option. The per-chromosome filtered outputs were then merged using plink2 --pmerge-list. 1,196,533 SNPs remained on the COMPLETE case after this step, and 467 for the COJO input. These were further pruned to remove highly-correlated SNPs using plink2's --indep-pairwise 200 50 0.25 option, where the values represent the window size, step size and maximum LD $r^2$ threshold allowed. A total of 1,002,886 and 8 SNPs were removed in this step on the COMPLETE and COJO cases, respectively.

The remaining SNPs were then employed to calculate the F coefficient for heterozygosity of each sample using plink2 --het. All individuals with F coefficients more than 3 standard deviations away from the overall mean were removed, leaving 3198 and 3199 samples in the COMPLETE and COJO cases, respectively. These were further pruned with plink2 --king-cutof 0.125 to avoid closely-related samples biasing the results, with 0.125 representing the relatedness level of second-degree relatives. To ensure reproducibility, a fixed seed value (1986) was also included. This pruning step left 2575 and 2502 individuals for the COMPLETE and COJO analyses, respectively. The final target data files were then generated with plink2 --make-bed specifying the remaining variants and individuals.

Clumping of the remaining variants – retaining only weakly-correlated SNPs most associated with the phenotype of interest – is not yet available on plink2, so plink --clump from version 1.9-170906 was employed instead, with parameters --clump-p1 1, --clump-r2 0.1, and --clump-kb 250. Whereas no significant clumps were identified for the COJO analysis, 1414 clumps from the top 2213 variants were generated for the COMPLETE input in this manner.

Poligenic risk scores (PRSs) were finally computed using plink2 --score at different p-value thresholds indicated with the --q-score-range option. The generated values at each threshold were then employed to construct linear models with rDNA CN estimates as independent variable, as well as the 6 first principal components calculated on the pre-clumping variants with plink2 --pca, sex and 1000 genomes population as covariates. The goodness of fit of each model was then estimated as the difference between the $R^2$ of the model itself minus the $R^2$ of a null model without the PRSs.

**Genome-wide association study for rDNA CN.** Association between SNVs and rDNA CN was performed using the plink2 --glm command from plink release 2.0-20200328, with parameters --mind 0.05, --geno 0.05, --maf 0.05, and --hwe 0.001 for pre-filtering.

**General statistical analysis.** Final figures and statistical analysis were performed using GraphPad Prism (v9.2.0) and R (v4.1.1). Specific statistical analyses are indicated in the respective figure and table legends. All tests are two-sided unless specified otherwise.

### Reporting summary

Further information on research design is available in the Nature Portfolio Reporting Summary linked to this article.

## Data availability

Whole genome bisufite sequencing data generated for this study will be available from The Sequence read Archive (Home - SRA - NCBI (nih.

gov)) using Bioproject ID: PRJNA817350 upon manuscript acceptance. External data sources were available in public repositories: GEO (GSE87893), (GSE157551), European Genome Archive (Dataset ID EGAD00001009789) and 1000 Genomes Project. According to the Danish and EU legislations, transfer and sharing of individual-level data require prior approval from the Danish Data Protection Agency and require that data sharing requests are dealt with on a case-by-case basis. For this reason, the raw data on the monozygotic twin cohort cannot be deposited in a public database but can be made available through collaboration or upon request (contact Qihua Tan, qtan@health.sdu.dk). All data was aligned to the GRCh38 assembly, unless otherwise specified. rDNA consensus sequences were obtained from GenBank (human KY962518.1, rat V01270.1).

## Code availability

All programs and parameters are described within the relevant methods sections, but clarification can be provided upon request.

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

## Acknowledgements

We thank Christopher Bell (QMUL) and Sue Ozanne (IMS, Cambridge) for their helpful comments on the manuscript. For the purpose of open access, the author has applied a Creative Commons Attribution (CC BY) licence to any Author Accepted Manuscript version arising. This work was supported by an Academy of Medical Sciences Springboard Award (SBF003\1026) (M.L.H.), Medical Research Council (MR/X009661/1) (M.L.H.), Medical Research Council (MR/P011799/1) (D.J.W., V.K.R., M.L.H., S.H.), Barts Charity Grant (MGU0604) (V.K.R.), Rosetrees grant (Seedcorn 2021\100182) (V.K.R.), Rosetrees grant (CF-2021-2\109) (V.K.R.), BBSRC grant (BB/R00675X/1) (V.K.R. & F.R.-A.). D.J.W.'s salary is partly supported by the National Institute for Health and Care Research University College London Hospitals Biomedical Research Centre.

## Author contributions

Conceptualization: M.L.H., V.K.R., D.J.W. Methodology: M.L.H., V.K.R., D.J.W., P.P.L., F.R.-A., L.A.M. Investigation: P.P.L., L.A.M., F.R.-A., F.A., M.G., M.L.H., R.E.A.S., R.M., M.N.A., W.L., Q.T., S.Y., J.R.C.M., H.T. Visualization: M.L.H., P.P.L., F.R.-A., L.A.M. Funding acquisition: M.L.H., V.K.R., D.J.W., S.L.H. Project administration: M.L.H., V.K.R., D.J.W. Supervision: M.L.H., V.K.R., D.J.W. Writing – original draft: M.L.H., P.P.L., F.R.-A. Writing – review & editing: M.L.H., V.K.R., F.R.-A., L.A.M., P.P.L., R.E.A.S., D.J.W. These authors contributed equally: P.P.O.L., L.A.M., F.R.-A.

## Competing interests

The authors declare no competing interests.
