## [Peer Review File · Nature Communications]

Ribosomal DNA copy number is associated with body mass in humans and other mammalsREVIEWER COMMENTS

Reviewer #1 (Remarks to the Author):

The article by Struijk and Rodriguez-Algarra et al. studied copy number and DNA methylation changes across the 45S rDNA in obese individuals. The authors used whole-genome approaches to determine the role of rDNA copy number/methylation in obesity in both humans and rats. Their study shows an association between rDNA copy number and DNA methylation with obesity in humans and an effect of germ-line inherited rDNA copy number on growth rates in rats. The findings will be of interest for researchers working on ribosomal DNA and will open new avenues towards understanding the role of the 45S rDNA locus in obesity. The results are novel particularly since the 45S rDNA sequence has been previously excluded from reference genomes because of its highly repetitive nature. Overall, the study is well designed and the results will be of significance to the field. Furthermore, the manuscript is well written and the results are well described.

I have the following suggestions/comments to help improve the manuscript:

1. The authors used WGBS and RRBS to determine rDNA copy number count, however the effect of depth coverage on the calculated rDNA counts has not been properly evaluated. Please consider checking the effect of depth coverage by reducing the read count of a single library by 90%, 80%, 70%, 60%, 50%, etc. before estimating rDNA CN in both the WGBS and RRBS libraries. Please also indicate whether a deduplication step was included in your pipeline for WGBS analysis. One would expect that a deduplication step would lead to reduced rDNA CN values in libraries with higher depth coverage due to the repetitive nature of the locus. The step is usually not applied in RRBS, however it is employed during the analysis of WGBS libraries.
2. In the first analysis, the authors only focused on male samples to determine if DNA methylation of repetitive elements is associated with BMI. Is there a specific reason as to why this analysis was restricted to males?
3. Please provide additional information regarding the regions used for normalizing the calculated rDNA counts. Are those regions included only on acrocentric chromosomes?
4. The authors tested whether rDNA copy number varies among different blood cell types. Based on their findings, they mentioned that the calculated rDNA CN association with BMI is not affected by blood cell composition across samples. Did the authors also check the degree of change in rDNA methylation across the different blood cell types? There was no correction employed for cell type composition, therefore it is important to determine whether rDNA methylation changes exist between the different cell types. This will help exclude cell type composition as a possible factor contributing to the differences in rDNA methylation between obese vs lean individuals.
5. In Figure S7, two donors have significant differences in rDNA CN between their cell types when compared to the other three donors. Did the authors check potential reasons for the rDNA CN differences in those individuals (Grey dots and Orange Dots) across the studied cell types?
6. The authors mentioned the following in their manuscript: "To the best of our knowledge, cell-type specific variation in rDNA CN has not been characterized in humans." Recently, a study has determined intra-individual variation in rDNA CN in different human tissues. The results from this article (PMID: 37368968) align with the authors' observations.
7. In Figure 1, I would suggest to include additional boxplots showing the average rDNA methylation separately across the upstream control element (UCE) and the core promoter.
8. The authors propose a scenario for explaining the origin of association between BMI and rDNA copy number. This scenario suggests that the somatic rDNA count is inherited through the germline and remains stable across all adult tissues. Based on this, I would suggest to additionally look into sperm WGBS/RRBS libraries to determine whether males with higher BMI have a distinct rDNA CN/methylation signature compared to lean males. It would be also interesting to check intra-individual variation in rDNA CN in multiple single sperm cells from the same male. Data from single sperm WGBS libraries are published as part of the following paper (PMID: 29255258), which might be used by the authors for this analysis. Similar publicly available datasets might be also available.

Reviewer #2 (Remarks to the Author):

This work from Law et al. describes the association between rDNA copy number (CN) and assessments of body size in both rats and humans, finding that lower rDNA CN is associated with higher weight and BMI. The authors additionally use analysis from monozygotic twin pairs to suggest that increased BMI is a consequence of lower inherited rDNA CN, and compare rDNA CN-weight associations in pre- and post-pubescent rats to demonstrate rDNA CN is negatively associated with pubescent weight gain. The implication of this work that normal physiological variations in rDNA CN has physiological consequences in mammals, including humans, is an important advancement for the field, directly implicating the biology directing rDNA CN with human health. However, these findings rely on accepting the accuracy of rDNA CN quantification at face value, and further validation of the authors' methods is essential to solidify the authors' analysis. Furthermore, some of the authors' claims are overstated or fail to recognize essential caveats, and the manuscript would be greatly improved by a more nuanced interpretation of the analyses.

Major comments

1) All of the major findings of this work are dependent on the accurate rDNA CN quantification calculated from whole genome sequencing methods, which vary between experiments depending on the data source. For this reason, it is critical that the authors demonstrate the accuracy of their quantification methods to substantiate these phenotype-rDNA CN associations. ddPCR has emerged as a highly accurate method of rDNA quantification, and the authors refer to their previous work to demonstrate a strong correlation between rDNA CN determined by ddPCR and WGBS and validate their rDNA CN quantification method (Rodriguez-Algarra et al., 2022). The correlation in this previous work however is weaker at values higher than 200 rDNA copies, suggesting biased underrepresentation of high rDNA CN by WGBS. Since the majority of the samples in the authors' WGBS analysis (Figure 1) have more than 200 rDNA copies, it is critical that the authors validate the accuracy of their WGBS rDNA quantification by ddPCR across the observed range of rDNA copies to ensure their observed effects are the result of true rDNA CN differences and not biases in quantification by WGBS.

2) The analyses in Figures 2-4 use existing RRBS data sets to quantify rDNA CN. The authors demonstrate an association of rDNA CN determined by RRBS and WGBS (Figure S4), although this association is somewhat weak. Importantly, no matter how strong the association between the two rDNA CN quantification methods (for either WGBS-ddPCR or WGBS-RRBS comparisons), the most important feature is the slope of the association, which should equal 1. Any association greater than or less than 1 indicates one of the methods has non-uniform quantification across the range of the data which would distort the difference in rDNA CN between samples. For example, the data in Figure S4 appears to have a slope greater than 1, indicating RRBS biasedly underrepresents rDNA CN at high values. For all methodological assessments, the authors should use statistical analysis to demonstrate that the slope of the associations is not significantly different than 1. If it is, the bias in rDNA CN quantification should be mentioned in the main text, especially when a lack of differences in rDNA CN is observed between conditions (such as in the twin analysis).

3) The lack of rDNA CN discordance in twin pairs with divergent BMI is intriguing and supports the authors suggestions that BMI does not influence rDNA CN, however the authors' statement that their analyses "support the idea that rDNA CN is somatically stable" is not supported by their data. While it appears that BMI does not influence rDNA CN, the existence of any rDNA CN differences between MZ twins indicates that rDNA CN is actually somatically unstable and this instability is influenced by environmental factors that differ between those twins. Furthermore, the similarity in rDNA CN across tissue types does not support that rDNA CN is somatically stable, only that any potential somatic

instability is not random, but is instead uniform across tissues (such as a consistent rate of rDNA CN loss over time or in response to specific environmental cues). The authors should edit the manuscript so their interpretations more accurately reflect the data.

Minor comments

The analysis of rDNA copy number during development in rats is very intriguing. Including the data from 8 week old rats in the main figure, rather than a summary of the analysis in table S12, would emphasize that the association between rDNA CN and weight is not observed until after puberty (week 19).

The analysis of rDNA CN from multiple tissues is not clear because it only provides summary statistics in a single table (Table S5). A figure displaying the data across all tissue types similar to Figure S7 would make the analysis more transparent for the readers.

Since Figure 3A-C represents paired data, the figures should reflect that with lines connecting the data between each "leaner" and "heavier" twin pair.

In response to the comments provided by reviewer #1:

- “the effect of depth coverage on the calculated rDNA counts has not been properly evaluated. Please consider checking the effect of depth coverage by reducing the read count of a single library by 90%, 80%, 70%, 60%, 50%, etc. before estimating rDNA CN in both the WGBS and RRBS libraries.”

The reviewer is indeed correct, estimation of rDNA copy number is very much affected by the depth of coverage, as has been highlighted in analyses by other groups previously¹ and mentioned within our original and revised manuscript¹⁻³. It is worth noting that in the previous and revised version of the manuscript, we also addressed that there was no systematic bias in sequencing depth that could account for the body mass associations that we report (now Figures S3, S9, S12, S16). Nonetheless, we have in addition performed the analyses suggested by the reviewer. These additional analyses are now included in the supplemental information (Fig. S21 and reproduced below) and described with an additional section in the methods, “QC of rDNA CN estimation” on p.20 of the revised manuscript. The figure is also shown below (Response Fig. 1). These analyses show that the variability of rDNA copy number estimates increases with reducing read coverage. Nonetheless, for each down-sampling level, the average copy number is close to the value estimated using the full dataset. Most of the libraries analysed have a coverage 100 million (WGBS) and 10 million (RRBS) reads, so the sequencing depth is not expected to have an effect on the reported results (Figures S3, S9, S12, S16, S21) for the datasets used in any of the association analyses.

Response Fig. 1. Read coverage downsampling affects rDNA copy number estimate variability in WGBS and RRBS datasets. High-coverage WGBS (A, B) and RRBS (C, D) samples are selected for downsampling at levels of 40%, 50%, 60%, 70%, 80% and 90%. At each downsampling level, 10 random downsamplings were performed. Decreasing read coverage increases the deviation of rDNA copy number estimates while the mean stays close to the value estimated in the full dataset (orange line).

- “Please also indicate whether a deduplication step was included in your pipeline for WGBS analysis.”

We compared performing a deduplication step in the WGBS data to not deduplicating and found no difference in the outcome for rDNA CN estimates (Response Fig. 2). In light of no effect on the results, we did not perform deduplication on the WGBS data reported in the manuscript to maintain as much consistency as possible across the various analyses as deduplication is not relevant for RRBS data. We have changed the wording within the “sequencing data processing” section of the methods in the revised version of the manuscript (p.19) to clarify this. Furthermore, to add transparency, we have included the results of this analysis for the reviewer below.

Response Fig. 2. Deduplication has no strong effect on rDNA copy number estimates in whole-genome bisulfite sequencing data. Deduplication was performed using `bismark_deduplication`. **(A)** Total library coverages of two selected WGBS samples (D454, black; D284, red) are reduced by ~20% after deduplication. **(B)** Deduplication reduced the number of reads uniquely aligned to rDNA by 26% in D454 and 29% in D284 samples. **(C)** Relative rDNA copy numbers are not affected by deduplication step.

- *“In the first analysis, the authors only focused on male samples to determine if DNA methylation of repetitive elements is associated with BMI. Is there a specific reason as to why this analysis was restricted to males?”*

We included only male samples in the initial WGBS analysis of BMI association as these were the samples which had been collected by our clinical collaborator when the collaboration for this study was initiated.

- *“Please provide additional information regarding the regions used for normalizing the calculated rDNA counts. Are those regions included only on acrocentric chromosomes?”*

Sequences were not limited to the acrocentric chromosomes for normalisation. Exons from the sex chromosomes and less than 300 bp were removed, as were sequences with high similarity to each other. This approach is consistent with what we and others have performed previously, which also have not limited normalisation regions to the acrocentric chromosomes^{1,3,4}. The relevant detail and citations are contained within the manuscript under the methods section, “rDNA CN estimation” (p.20 revised manuscript).

- *“calculated rDNA CN association with BMI is not affected by blood cell composition across samples. Did the authors also check the degree of change in rDNA methylation across the different blood cell types? There was no correction employed for cell type composition, therefore it is important to determine whether rDNA methylation changes exist between the different cell types.”*

In the previous version of the manuscript, we did not look at rDNA methylation across the sorted blood cell groups. The data presented throughout our manuscript suggests that the level of DNA methylation in adult tissues is determined in response to the rDNA copy number, therefore, for clarity we did not muddy the waters by including additional, less relevant analyses that may obscure the conclusion that underlying genetic variation at rDNA is what is driving the association. Furthermore, we and others have previously shown that there does not appear to be systematic difference in the rDNA CN across tissues² or individual cell types (Fig. S11) from a given individual.

- *“In Figure S7, two donors have significant differences in rDNA CN between their cell types when compared to the other three donors. Did the authors check potential reasons for the rDNA CN differences in those individuals (Grey dots and Orange Dots) across the studied cell types?”*

The higher variation in estimates of rDNA copy numbers in this dataset is due to data of lower quality being employed for this analysis compared to the filters applied to all data used in any of the association analyses. Specifically, the bisulfite conversion efficiency is less compared to the other data and also the mapping efficiency is reduced. Therefore, this is a noisy dataset. However, it is sufficiently robust to demonstrate the point that there is no systematic cell type specific variation in rDNA copy number as concluded in the original version of the manuscript and represented in the revised manuscript, Fig. S11 ($p_{\text{donor}} = 3.8 \times 10^{-11}$, $p_{\text{cell_type}} = 0.8604$), together with what has been independently reported elsewhere².

- *“The authors mentioned the following in their manuscript: “To the best of our knowledge, cell-type specific variation in rDNA CN has not been characterized in humans.” Recently, a study has determined intra-individual variation in rDNA CN in different human tissues. The results from this article (PMID: 37368968) align with the authors’ observations.”*

We thank the reviewer for drawing our attention to this recent publication. Indeed, the authors validate similarly to here and our previous work³ that WGBS can be used similarly to WGS to estimate rDNA copy numbers, with cross validation of the methods using the 18S rDNA subunit producing correlation coefficients > 0.9. The findings reported in their manuscript strongly support our own observations that rDNA copy number and methylation are highly correlated and also that there is little evidence of copy number variation between tissues of an individual². This further supports the analyses based on sorted cell populations discussed in the point above. Revised text is as follows, *“rDNA CN was estimated from the WGBS data using a method based on that for WGS⁴. We and others have previously cross validated this approach^{2,3} and also applied strict criteria for data quality control, as this can influence the accuracy of CN assessment¹ (Table S3, Fig. S3).”*-p4 of revised manuscript, and, *“However, total rDNA CN has previously been shown to be consistent across multiple tissues in mice⁵ and more recently humans^{2”}*-p 7 of revised manuscript.

- *“In Figure 1, I would suggest to include additional boxplots showing the average rDNA methylation separately across the upstream control element (UCE) and the core promoter.”*

We appreciate the reviewers’ suggestion to provide an additional representation of the regulatory regions of the 47S-rDNA and have now added the requested plots as a supplemental figure (Fig. S2). Methylation across these regulatory regions is similarly hypomethylated in obese individuals as was observed when considering the entire transcribed unit and 1000 bp upstream. This figure has been reproduced below for ease of reference (Response Fig. 3).

Response Fig.3. Methylation of core regulatory units within the rDNA promoter. (A) Positions of the upstream control element (UCE) and core promoter (CORE) relative to the transcriptional start site (+1) are indicated. **(B)** Methylation of the UCE is significantly lower in the obese (n=32) compared to the lean (n=31) group (Mann-Whitney test, p=0.0213). **(C)** Methylation of the CORE is significantly lower in the obese (n=32) compared to the lean (n=31) group (Mann-Whitney test, p=0.0177).

- *"I would suggest to additionally look into sperm WGBS/RRBS libraries to determine whether males with higher BMI have a distinct rDNA CN/methylation signature compared to lean males. It would be also interesting to check intra-individual variation in rDNA CN in multiple single sperm cells from the same male. Data from single sperm WGBS libraries are published as part of the following paper (PMID: 29255258), which might be used by the authors for this analysis."*

The reviewer poses interesting analyses to address whether the association between rDNA copy number in somatic tissues with adult body mass is preserved in the germline. Unfortunately, we do not have matched sperm/somatic tissue data to be able to address this question in humans. However, our previously published work has shown that there is a strong correlation between rDNA promoter methylation levels between liver tissue and sperm in adult mice (Figure S4) in Holland *et al.*,⁶. Given the high level of correlation between rDNA copy number and methylation, we therefore suggest that it is likely that bulk sperm populations will show similar inter-individual differences in copy number to that observed in somatic tissue. The reviewer further suggests using single-cell WGBS data to address this question. Although this would be a very interesting study in its own right, unfortunately single-cell data suffers from patchy sampling across the genome which means that estimation of rDNA copy number normalised to exon reads is not feasible at present⁷. Based on the evidence that suggests that rDNA represents a part of the genome that experiences an accelerated rate of recombination in

meiosis^{8,9}, leading to the copy number loss and gain thought to underlie the high amount of interindividual copy number variation, it would be very interesting to see direct evidence of this using a single cell analysis when technology evolves to permit this. I would expect to see single cell variation resulting from meiotic recombination.

In response to the comments provided by reviewer #2:

- *“it is critical that the authors validate the accuracy of their WGBS rDNA quantification by ddPCR across the observed range of rDNA copies to ensure their observed effects are the result of true rDNA CN differences and not biases in quantification by WGBS.”*

It is important to note that in the original version of the manuscript, biases in quantification methods are unlikely to contribute to any of the observed associations, as there were strong quality control measures employed across each data set to ensure limited data variability within a given dataset (revised manuscript, Fig. S3, S9, S12, S16) and further supported by the down-sampling analysis requested in the first point by Reviewer 1 above (revised manuscript, Fig. S21). Data was never combined from different sources and analyses were performed to ensure that any variation in data quality within a given dataset was not confounded with any physiological measurements, including body weight related parameters. Furthermore, in the original and revised manuscripts we do not attempt to define definitive rDNA copy number ranges with a specific risk ratio for higher or lower body mass, rather it is a relative association which we report in this discovery study. Indeed, for clinical translation as a genetic risk factor, a more quantitative association would be required, but this is beyond the scope of this foundational study.

In addition to highlighting the above, we also performed ddPCR as requested on the original samples for which the WGBS analysis was performed where we had enough sample remaining, obtaining data, after excluding samples with Lambda values ≥ 1.6 for 48/63 samples included in the WGBS analysis. We have now included this additional analysis as supplementary Figure S3 and modified the methods and main manuscript to describe these results accordingly. This figure is included below for easy reference (Response Figure 4), demonstrating technical replication of the findings using an independent methodology.

Response Figure 4. Validation of WGBS rDNA CN assessment by a sequencing independent method-digital droplet PCR (ddPCR). A) rDNA CN determined by WGBS is correlated with

rDNA CN assessed by ddPCR (Spearman $r = 0.8135$, $p < 0.0001$, $n = 48$). **B**) rDNA CN in the obese group is lower when assessed using ddPCR ($p = 0.0492$, Mann-Whitney test). **C**) rDNA CN in the obese group is lower when assessed using WGBS ($p = 0.0119$, Mann-Whitney test). Throughout lean ($BMI < 25 \text{ kg/m}^2$, $n=19$, black) or obese ($BMI > 30 \text{ kg/m}^2$, $n=29$, red). Sample number is restricted due to not having sufficient amount of sample remaining to perform ddPCR validation for all individuals that passed the WGBS QC.

- *“Importantly, no matter how strong the association between the two rDNA CN quantification methods (for either WGBS-ddPCR or WGBS-RRBS comparisons), the most important feature is the slope of the association, which should equal 1.... For all methodological assessments, the authors should use statistical analysis to demonstrate that the slope of the associations is not significantly different than 1. If it is, the bias in rDNA CN quantification should be mentioned in the main text, especially when a lack of differences in rDNA CN is observed between conditions (such as in the twin analysis).”*

We appreciate the reviewers' comment. Indeed, any given scientific methodology will have some limitations and any experimental method will generate data with some element of noise. To reassure the reviewer we have now further assessed for any systematic biases in our approaches, we have performed the requested statistical analyses. The slope of the WGBS and RRBS data was not significantly different for 1 (Student t-test, $p = 0.9776$) when the RRBS units were rescaled to the same base units as in WGBS (10^3), the slope of linear association is estimated as 1.0084 ± 0.2841 (S.D.). We therefore, do not believe that this is a confounding technical factor that could explain the lack of association observed in the MZ twin data. Slope of the linear association between WGBS and ddPCR is estimated as 0.8335 ± 0.0912 (S.D.). Statistical analysis demonstrated that there is not enough evidence from the data to suggest that slope of association is statistically different from 1 (Student t test, $p = 0.0744$).

- *“The lack of rDNA CN discordance in twin pairs with divergent BMI is intriguing and supports the authors suggestions that BMI does not influence rDNA CN, however the authors' statement that their analyses “support the idea that rDNA CN is somatically stable” is not supported by their data. While it appears that BMI does not influence rDNA CN, the existence of any rDNA CN differences between MZ twins indicates that rDNA CN is actually somatically unstable and this instability is influenced by environmental factors that differ between those twins. Furthermore, the similarity in rDNA CN across tissue types does not support that rDNA CN is somatically stable, only that any potential somatic instability is not random, but is instead uniform across tissues (such as a consistent rate of rDNA CN loss over time or in response to specific environmental cues). The authors should edit the manuscript so their interpretations more accurately reflect the data.”*

We thank the reviewer for their comment. We clarify that there is no significant rDNA CN differences between the MZ twins as indicated in Fig. 3B (Wilcoxon matched-pairs signed rank test, $p = 0.8115$) and is significantly paired (Spearman $r = 0.9670$, $p < 0.0001$), suggesting that any somatic instability in “healthy” individuals is limited. Furthermore, another recent study has also provided additional verification for little evidence of tissue-specific variation in rDNA CN², in addition to the analyses contained we have performed on purified cell populations (Fig. S11, Type II ANOVA; $p_{\text{donor}} = 3.8 \times 10^{-11}$, $p_{\text{cell_type}} = 0.8604$). Nonetheless, we do appreciate the point that this is not conclusive evidence that rDNA copy number may not systematically change across all tissues in response to environmental cues, especially *in utero* due to early environmental factors that would have also been shared by the MZ

twins. Therefore, we have now changed the wording to, “These results, together with the cell type specific analysis above, support the idea that rDNA CN is not subject to extensive random drift over time or in response to environmental or metabolic changes associated with altered BMI. It is therefore more likely that it is germline inherited rDNA CN that is associated with adult BMI,”(p8, revised manuscript) to reflect that there is a level of uncertainty around this interpretation.

- “The analysis of rDNA copy number during development in rats is very intriguing. Including the data from 8 week old rats in the main figure, rather than a summary of the analysis in table S12, would emphasize that the association between rDNA CN and weight is not observed until after puberty (week 19).”

We have followed the reviewers’ advice and have reformatted Figure 4 in the revised manuscript to now include the data from the 8 week old rats (Response Fig.4). We also deleted the previous supplementary table.

Response Fig. 4. Weight gain from puberty to early adulthood negatively correlates with rDNA copy number in Sprague-Dawley rats. RRBS data derived from liver of female Sprague-Dawley rats at the time of sacrifice. **(A)** Weekly weight data fits an exponential plateau curve ($R^2=0.7232$). **(B)** Absolute weight is not negatively correlated with rDNA copy number at week 8 (Spearman $r = -0.07181$, $p=0.6432$, $n=44$). **(C)** Absolute weight is negatively correlated with rDNA copy number at week 19 (Spearman $r = -0.3481$, $p=0.0206$, $n=44$). **(D)** Weight gain (week 8-19) is negatively correlated with rDNA copy number (Spearman $r = -0.4116$, $p=0.0055$, $n=44$).

- “The analysis of rDNA CN from multiple tissues is not clear because it only provides summary statistics in a single table (Table S5). A figure displaying the data across all tissue types similar to Figure S7 would make the analysis more transparent for the readers.”

A limitation of this data source is that common donors are only available for multiple cell types from a single tissue origin. We originally only showed the graphic data for the blood cell types as this is the

most relevant data to show that there is no systematic copy number changes in specific white blood cell types that could influence the association between rDNA copy number and BMI found in blood (discovery cohort) or adipose tissue (validation cohort). Nonetheless, we have included the other tissue types and updated the revised manuscript with to include Fig. S11, also shown below (Response Fig. 5).

Response Fig.5. rDNA CN in purified cell populations. Different donors are indicated by the different colour/symbol combinations. Type II ANOVA shows that when all cell types are considered, only donor origin influences rDNA CN ($p_{\text{donor}} = 3.8 \times 10^{-11}$, $p_{\text{cell_type}} = 0.8604$). This is also true when only isolated blood cell types are included ($p_{\text{donor}} = 7.5 \times 10^{-8}$, $p_{\text{cell_type}} = 0.8414$).

- “Since Figure 3A-C represents paired data, the figures should reflect that with lines connecting the data between each “leaner” and “heavier” twin pair.”

We have included this format (Response Fig.6).

Figure 3. BMI does not induce rDNA copy number variation and is not influenced by BMI-associated genetic variation in the rest of the genome. BMI clinical classifications are indicated (lean (BMI < 25 kg/m²) black, overweight (25 kg/m² < BMI < 30 kg/m²) blue, obese (BMI > 30 kg/m²) red). Sex is indicated by symbol shape, circle = female (14 MZ twin pairs), square = male (10 MZ twin pairs). **(A)** BMI is discordant between twins (Wilcoxon matched-pairs signed rank test, $p < 0.0001$) and significantly paired (Spearman $r = 0.8723$, $p < 0.0001$). **(B)** Relative rDNA copy number is not different between twins (Wilcoxon matched-pairs signed rank test, $p = 0.8115$) and is significantly paired (Spearman $r = 0.9670$, $p < 0.0001$). **(C)** rDNA methylation is not different between twins (Wilcoxon matched-pairs signed rank test, $p = 0.7048$) and is significantly paired (Spearman $r = 0.9783$, $p < 0.0001$). **(D)** BMI polygenic risk scores are not correlated with rDNA copy number (Spearman $r = 0.01284$, $p = 0.5304$, $n = 2390$).

References

- 1 Hall, A. N., Turner, T. N. & Queitsch, C. Thousands of high-quality sequencing samples fail to show meaningful correlation between 5S and 45S ribosomal DNA arrays in humans. *Sci Rep* **11**, 449 (2021). <https://doi.org:10.1038/s41598-020-80049-y>
- 2 Razzaq, A., Bejaoui, Y., Alam, T., Saad, M. & El Hajj, N. Ribosomal DNA Copy Number Variation is Coupled with DNA Methylation Changes at the 45S rDNA Locus. *Epigenetics* **18**, 2229203 (2023). <https://doi.org:10.1080/15592294.2023.2229203>
- 3 Rodriguez-Algarra, F. *et al.* Genetic variation at mouse and human ribosomal DNA influences associated epigenetic states. *Genome Biol* **23**, 54 (2022). <https://doi.org:10.1186/s13059-022-02617-x>
- 4 Gibbons, J. G., Branco, A. T., Godinho, S. A., Yu, S. & Lemos, B. Concerted copy number variation balances ribosomal DNA dosage in human and mouse genomes. *Proc Natl Acad Sci U S A* **112**, 2485-2490 (2015). <https://doi.org:10.1073/pnas.1416878112>
- 5 Xu, B. *et al.* Ribosomal DNA copy number loss and sequence variation in cancer. *PLoS Genet* **13**, e1006771 (2017). <https://doi.org:10.1371/journal.pgen.1006771>
- 6 Holland, M. L. *et al.* Early-life nutrition modulates the epigenetic state of specific rDNA genetic variants in mice. *Science* **353**, 495-498 (2016). <https://doi.org:10.1126/science.aaf7040>
- 7 Smallwood, S. A. *et al.* Single-cell genome-wide bisulfite sequencing for assessing epigenetic heterogeneity. *Nat Methods* **11**, 817-820 (2014). <https://doi.org:10.1038/nmeth.3035>
- 8 Guarracino, A. *et al.* Recombination between heterologous human acrocentric chromosomes. *Nature* **617**, 335-343 (2023). <https://doi.org:10.1038/s41586-023-05976-y>
- 9 Stults, D. M., Killen, M. W., Pierce, H. H. & Pierce, A. J. Genomic architecture and inheritance of human ribosomal RNA gene clusters. *Genome Res* **18**, 13-18 (2008). <https://doi.org:10.1101/gr.6858507>

Original reviewers' comments in full:

Reviewer #1 (Remarks to the Author):

The article by Struijk and Rodriguez-Algarra et al. studied copy number and DNA methylation changes across the 45S rDNA in obese individuals. The authors used whole-genome approaches to determine the role of rDNA copy number/methylation in obesity in both humans and rats. Their study shows an association between rDNA copy number and DNA methylation with obesity in humans and an effect of germ-line inherited rDNA copy number on growth rates in rats. The findings will be of interest for researchers working on ribosomal DNA and will open new avenues towards understanding the role of the 45S rDNA locus in obesity. The results are novel particularly since the 45S rDNA sequence has been previously excluded from reference genomes because of its highly repetitive nature. Overall, the study is well designed and the results will be of significance to the field. Furthermore, the manuscript is well written and the results are well described.

I have the following suggestions/comments to help improve the manuscript:

1. The authors used WGBS and RRBS to determine rDNA copy number count, however the effect of depth coverage on the calculated rDNA counts has not been properly evaluated. Please consider checking the effect of depth coverage by reducing the read count of a single library by 90%, 80%, 70%, 60%, 50%, etc. before estimating rDNA CN in both the WGBS and RRBS libraries. Please also indicate whether a deduplication step was included in your pipeline for WGBS analysis. One would expect that a deduplication step would lead to reduced rDNA CN values in libraries with higher depth

coverage due to the repetitive nature of the locus. The step is usually not applied in RRBS, however it is employed during the analysis of WGBS libraries.

2. In the first analysis, the authors only focused on male samples to determine if DNA methylation of repetitive elements is associated with BMI. Is there a specific reason as to why this analysis was restricted to males?

3. Please provide additional information regarding the regions used for normalizing the calculated rDNA counts. Are those regions included only on acrocentric chromosomes?

4. The authors tested whether rDNA copy number varies among different blood cell types. Based on their findings, they mentioned that the calculated rDNA CN association with BMI is not affected by blood cell composition across samples. Did the authors also check the degree of change in rDNA methylation across the different blood cell types? There was no correction employed for cell type composition, therefore it is important to determine whether rDNA methylation changes exist between the different cell types. This will help exclude cell type composition as a possible factor contributing to the differences in rDNA methylation between obese vs lean individuals.

5. In Figure S7, two donors have significant differences in rDNA CN between their cell types when compared to the other three donors. Did the authors check potential reasons for the rDNA CN differences in those individuals (Grey dots and Orange Dots) across the studied cell types?

6. The authors mentioned the following in their manuscript: "To the best of our knowledge, cell-type specific variation in rDNA CN has not been characterized in humans." Recently, a study has determined intra-individual variation in rDNA CN in different human tissues. The results from this article (PMID: 37368968) align with the authors' observations.

7. In Figure 1, I would suggest to include additional boxplots showing the average rDNA methylation separately across the upstream control element (UCE) and the core promoter.

8. The authors propose a scenario for explaining the origin of association between BMI and rDNA copy number. This scenario suggests that the somatic rDNA count is inherited through the germline and remains stable across all adult tissues. Based on this, I would suggest to additionally look into sperm WGBS/RRBS libraries to determine whether males with higher BMI have a distinct rDNA CN/methylation signature compared to lean males. It would be also interesting to check intra-individual variation in rDNA CN in multiple single sperm cells from the same male. Data from single sperm WGBS libraries are published as part of the following paper (PMID: 29255258), which might be used by the authors for this analysis. Similar publicly available datasets might be also available.

Reviewer #2 (Remarks to the Author):

This work from Law et al. describes the association between rDNA copy number (CN) and assessments of body size in both rats and humans, finding that lower rDNA CN is associated with higher weight and BMI. The authors additionally use analysis from monozygotic twin pairs to suggest that increased BMI is a consequence of lower inherited rDNA CN, and compare rDNA CN-weight associations in pre- and post-pubescent rats to demonstrate rDNA CN is negatively associated with pubescent weight gain. The implication of this work that normal physiological variations in rDNA CN has physiological consequences in mammals, including humans, is an important advancement for the field, directly implicating the biology directing rDNA CN with human health. However, these findings rely on accepting the accuracy of rDNA CN quantification at face value, and further validation of the authors' methods is essential to solidify the authors' analysis. Furthermore, some of the authors' claims are overstated or fail to recognize essential caveats, and the manuscript would be greatly improved by a more nuanced interpretation of the analyses.

Major comments

1) All of the major findings of this work are dependent on the accurate rDNA CN quantification calculated from whole genome sequencing methods, which vary between experiments depending on the data source. For this reason, it is critical that the authors demonstrate the accuracy of their quantification methods to substantiate these phenotype-rDNA CN associations. ddPCR has emerged as a highly accurate method of rDNA quantification, and the authors refer to their previous work to demonstrate a strong correlation between rDNA CN determined by ddPCR and WGBS and validate their rDNA CN quantification method (Rodriguez-Algarra et al., 2022). The correlation in this previous work however is weaker at values higher than 200 rDNA copies, suggesting biased underrepresentation of high rDNA CN by WGSB. Since the majority of the samples in the authors' WGBS analysis (Figure 1) have more than 200 rDNA copies, it is critical that the authors validate the accuracy of their WGBS rDNA quantification by ddPCR across the observed range of rDNA copies to ensure their observed effects are the result of true rDNA CN differences and not biases in quantification by WGBS.

2) The analyses in Figures 2-4 use existing RRBS data sets to quantify rDNA CN. The authors demonstrate an association of rDNA CN determined by RRBS and WGBS (Figure S4), although this association is somewhat weak. Importantly, no matter how strong the association between the two rDNA CN quantification methods (for either WGBS-ddPCR or WGBS-RRBS comparisons), the most important feature is the slope of the association, which should equal 1. Any association greater than or less than 1 indicates one of the methods has non-uniform quantification across the range of the data which would distort the difference in rDNA CN between samples. For example, the data in Figure S4 appears to have a slope greater than 1, indicating RRBS biasedly underrepresents rDNA CN at high values. For all methodological assessments, the authors should use statistical analysis to demonstrate that the slope of the associations is not significantly different than 1. If it is, the bias in rDNA CN quantification should be mentioned in the main text, especially when a lack of differences in rDNA CN is observed between conditions (such as in the twin analysis).

3) The lack of rDNA CN discordance in twin pairs with divergent BMI is intriguing and supports the authors suggestions that BMI does not influence rDNA CN, however the authors' statement that their analyses "support the idea that rDNA CN is somatically stable" is not supported by their data. While it appears that BMI does not influence rDNA CN, the existence of any rDNA CN differences between MZ twins indicates that rDNA CN is actually somatically unstable and this instability is influenced by environmental factors that differ between those twins. Furthermore, the similarity in rDNA CN across tissue types does not support that rDNA CN is somatically stable, only that any potential somatic instability is not random, but is instead uniform across tissues (such as a consistent rate of rDNA CN loss over time or in response to specific environmental cues). The authors should edit the manuscript so their interpretations more accurately reflect the data.

Minor comments

The analysis of rDNA copy number during development in rats is very intriguing. Including the data from 8 week old rats in the main figure, rather than a summary of the analysis in table S12, would emphasize that the association between rDNA CN and weight is not observed until after puberty (week 19).

The analysis of rDNA CN from multiple tissues is not clear because it only provides summary statistics in a single table (Table S5). A figure displaying the data across all tissue types similar to Figure S7 would make the analysis more transparent for the readers.

Since Figure 3A-C represents paired data, the figures should reflect that with lines connecting the data between each “leaner” and “heavier” twin pair.

REVIEWERS' COMMENTS

Reviewer #1 (Remarks to the Author):

Thank you for addressing all my comments in the revised version of the manuscript and for performing additional analysis to determine the effect of depth coverage and deduplication on the calculated rDNA counts.

Reviewer #2 (Remarks to the Author):

The authors were very thorough in their revisions, and fully addressed concerns with methodology and clarifying data transparency. The revised manuscript provides substantial support for the authors' findings.